# Multi-Information Source Optimization

**Matthias Poloczek**
Department of Systems and Industrial Engineering
University of Arizona
Tucson, AZ 85721
poloczek@email.arizona.edu

**Jialei Wang**
Chief Analytics Office
IBM
Armonk, NY 10504
jw865@cornell.edu

**Peter I. Frazier**
School of Operations Research and Information Engineering
Cornell University
Ithaca, NY 14853
pf98@cornell.edu

## Abstract

We consider Bayesian methods for multi-information source optimization (MISO), in which we seek to optimize an expensive-to-evaluate black-box objective function while also accessing cheaper but biased and noisy approximations ("information sources"). We present a novel algorithm that outperforms the state of the art for this problem by using a Gaussian process covariance kernel better suited to MISO than those used by previous approaches, and an acquisition function based on a one-step optimality analysis supported by efficient parallelization. We also provide a novel technique to guarantee the asymptotic quality of the solution provided by this algorithm. Experimental evaluations demonstrate that this algorithm consistently finds designs of higher value at less cost than previous approaches.

## 1 Introduction

We consider Bayesian multi-information source optimization (MISO), in which we optimize an expensive-to-evaluate black-box objective function while optionally accessing cheaper biased noisy approximations, often referred to as "information sources (IS)". This arises when tuning machine learning algorithms: instead of using the whole dataset for the hyperparameter optimization, one may use a small subset or even a smaller related dataset [34, 15, 17]. We also face this problem in robotics: we can evaluate a parameterized robot control policy in simulation, in a laboratory, or in a field test [15]. Cheap approximations promise a route to tractability, but bias and noise complicate their use. An unknown bias arises whenever a computational model incompletely models a real-world phenomenon, and is pervasive in applications.

We present a novel algorithm for this problem, `misoKG`, that is tolerant to both noise and bias and improves substantially over the state of the art. Specifically, our contributions are:

- The algorithm uses a novel acquisition function that maximizes the incremental gain per unit cost. This acquisition function generalizes and parallelizes previously proposed knowledge-gradient methods for single-IS Bayesian optimization [7, 8, 28, 26, 37] to MISO.

- We prove that this algorithm provides an asymptotically near-optimal solution. If the search domain is finite, this result establishes the consistency of `misoKG`.

  We present a novel proof technique that yields an elegant, short argument and is thus of independent interest.

**Related Work:**    To our knowledge, MISO was first considered by Swersky, Snoek, and Adams [34], under the the name multi-task Bayesian optimization. This name was used to suggest problems in which the auxiliary tasks could meaningfully be solved on their own, while we use the term MISO to indicate that the IS may be useful only in support of the primary task. Swersky et al. [34] showed that hyperparameter tuning in classification can be accelerated through evaluation on subsets of the validation data. They proposed a GP model to jointly model such "auxiliary tasks" and the primary task, building on previous work on GP regression for multiple tasks in [3, 10, 35]. They choose points to sample via cost-sensitive entropy search [11, 39], sampling in each iteration a point that maximally reduces uncertainty in the optimum's location, normalized by the query cost.

We demonstrate in experiments that our approach improves over the method of Swersky et al. [34], and we believe this improvement results from two factors: first, our statistical model is more flexible in its ability to model bias that varies across the domain; second, our acquisition function directly and maximally reduces simple regret in one step, unlike predictive entropy search which maximally reduces the maximizer's entropy in one step and hence only indirectly reduces regret.

Lam, Allaire, and Willcox [18] also consider MISO, under the name non-hierarchical multi-fidelity optimization. They propose a statistical model that maintains a separate GP for each IS, and fuse them via the method of Winkler [40]. They apply a modified expected improvement acquisition function on these surrogates to first decide what design $x^*$ to evaluate and then select the IS to query; the latter is decided by a heuristic that aims to balance information gain and query cost. We demonstrate in experiments that our approach improves over the method of Lam et al. [18], and we believe this improvement results from two factors: first, their statistical approach assumes an independent prior on each IS, despite their being linked through modeling a common objective; and second their acquisition function selects the point to sample and the IS to query separately via a heuristic rather than jointly using an optimality analysis.

Beyond these two works, the most closely related work is in the related problem of multi-fidelity optimization. In this problem, IS are supposed to form a strict hierarchy [16, 14, 6, 24, 20, 19, 15]. In addition, most of these models limit the information that can be obtained from sources of lower fidelity [16, 14, 6, 20, 19]: Given the observation of $x$ at some IS $\ell$, one cannot learn more about the value of $x$ at IS with higher fidelity by querying IS $\ell$ anywhere else (see Sect. C for details and a proof). Picheny et al. [24] propose a quantile-based criterion for optimization of stochastic simulators, supposing that all simulators provide *unbiased* approximations of the true objective. From this body of work, we compare against Kandasamy et al. [15], who present an approach for minimizing both simple and cumulative regret, under the assumption that the maximum bias of an information source strictly decreases with its fidelity.

An interesting special case of MISO is warm-starting Bayesian optimization. Here information sources correspond to samples that were taken previously on objective functions related to the current objective. For example, this scenario occurs when we are to re-optimize whenever parameters of the objective change or whenever new data becomes available. Poloczek et al. [25] demonstrated that a variant of the algorithm proposed in this article can reduce the optimization costs significantly by warm-starting Bayesian optimization, as does the algorithm of Swersky et al. [34].

Outside of both the MISO and multi-fidelity settings, Klein et al. [17] considered hyperparameter optimization of machine learning algorithms over a large dataset $D$. Supposing access to subsets of $D$ of arbitrary sizes, they show how to exploit regularity of performance across dataset sizes to significantly speed up the optimization process for support vector machines and neural networks.

Our acquisition function is a generalization of the knowledge-gradient policy of Frazier, Powell, and Dayanik [8] to the MISO setting. This generalization requires extending the one-step optimality analysis used to derive the KG policy in the single-IS setting to account for the impact of sampling a cheap approximation on the marginal GP posterior on the primary task. From this literature, we leverage an idea for computing the expectation of the maximum of a collection of linear functions of a normal random variable, and propose a parallel algorithm to identify and compute the required components.

The class of GP covariance kernels we propose are a subset of the class of linear models of coregionalization kernels [10, 2], with a restricted form derived from a generative model particular to MISO. Focusing on a restricted class of kernels designed for our application supports accurate inference with less data, which is important when optimizing expensive-to-evaluate functions.

Our work also extends the knowledge-gradient acquisition function to the variable cost setting. Similar extensions of expected improvement to the variable cost setting can be found in Snoek et al. [32] (the EI per second criterion) and in Le Gratiet and Cannamela [19].

We now formalize the problem we consider in Sect. 2, describe our statistical analysis in Sect. 3.1, specify our acquisition function and parallel computation method in Sects. 3.2 and 3.3, provide a theoretical guarantee in Sect. 3.4, present numerical experiments in Sect. 4, and conclude in Sect. 5.

## 2 Problem Formulation

Given a continuous objective function $g : \mathcal{D} \to \mathbb{R}$ on a compact set $\mathcal{D} \subset \mathbb{R}^d$ of feasible designs, our goal is to find a design with objective value close to $\max_{x \in \mathcal{D}} g(x)$. We have access to $M$ possibly biased and/or noisy IS indexed by $\ell \in [M]_0$. (Here, for any $a \in \mathbb{Z}^+$ we use $[a]$ as a shorthand for the set $\{1, 2, \ldots, a\}$, and further define $[a]_0 = \{0, 1, 2, \ldots, a\}$.) Observing IS $\ell$ at design $x$ provides independent, conditional on $f(\ell, x)$, and normally distributed observations with mean $f(\ell, x)$ and finite variance $\lambda_\ell(x)$. In [34], IS $\ell \in [M]_0$ are called "auxiliary tasks" and $g$ the primary task. These sources are thought of as approximating $g$, with variable bias. We suppose that $g$ can be observed directly without bias (but possibly with noise) and set $f(0, x) = g(x)$. The bias $f(\ell, x) - g(x)$ is also referred to as "model discrepancy" in the engineering and simulation literature [1, 4]. Each IS $\ell$ is also associated with a query cost function $c_\ell(x) : \mathcal{D} \to \mathbb{R}^+$. We assume that the cost function $c_\ell(x)$ and the variance function $\lambda_\ell(x)$ are both known and continuously differentiable (over $\mathcal{D}$). In practice, these functions may either be provided by domain experts or may be estimated along with other model parameters from data (see Sect. 4 and B.2, and [27]).

## 3 The `misoKG` Algorithm

We now present the `misoKG` algorithm and describe its two components: a MISO-focused statistical model in Sect. 3.1; and its acquisition function and parallel computation in Sect. 3.2. Sect. 3.3 summarizes the algorithm and Sect. 3.4 provides a theoretical performance guarantee. Extensions of the algorithm are discussed in Sect. D.

### 3.1 Statistical Model

We now describe a generative model for $f$ that results in a Gaussian process prior on $f$ with a parameterized class of mean functions $\mu : [M] \times \mathcal{D} \mapsto \mathbb{R}$ and covariance kernels $\Sigma : ([M] \times \mathcal{D})^2 \mapsto \mathbb{R}$. This allows us to use standard tools for Gaussian process inference — first finding the MLE estimate of the parameters indexing this class, and then performing Gaussian process regression using the selected mean function and covariance kernel — while also providing better estimates for MISO than would a generic multi-output GP regression kernel that does not consider the MISO application.

We construct our generative model as follows. For each $\ell > 0$ suppose that a function $\delta_\ell : \mathcal{D} \mapsto \mathbb{R}$ for each $\ell > 0$ was drawn from a separate independent GP, $\delta_\ell \sim GP(\mu_\ell, \Sigma_\ell)$, and let $\delta_0$ be identically 0. In our generative model $\delta_\ell$ will be the bias $f(\ell, x) - g(x)$ for IS $\ell$. We additionally set $\mu_\ell(x) = 0$ to encode a lack of a strong belief on the direction of the bias. (If one had a strong belief that an IS is consistently biased in one direction, one may instead set $\mu_\ell$ to a constant estimated using maximum a posteriori estimation.) Next, within our generative model, we suppose that $g : \mathcal{D} \mapsto \mathbb{R}$ was drawn from its own independent GP, $g \sim GP(\mu_0, \Sigma_0)$, for some given $\mu_0$ and $\Sigma_0$, and suppose $f(\ell, x) = f(0, x) + \delta_\ell(x)$ for each $\ell$. We assume that $\mu_0$ and $\Sigma_\ell$ with $\ell \geq 0$ belong to one of the standard parameterized classes of mean functions and covariance kernels, e.g., constant $\mu_0$ and Matérn $\Sigma_\ell$.

With this construction, $f$ is a GP: given any finite collection of points $\ell_i \in [M], x_i \in \mathcal{D}$ with $i = 1, \ldots, I$, $(f(\ell_i, x_i) : i = 1, \ldots, I)$ is a sum of independent multivariate normal random vectors, and thus is itself multivariate normal. Moreover, we compute the mean function and covariance kernel of $f$: for $\ell, m \in [M]_0$ and $x, x' \in \mathcal{D}$,

$$\mu(\ell, x) = \mathbb{E}\left[f(\ell, x)\right] = \mathbb{E}\left[g(x)\right] + \mathbb{E}\left[\delta_\ell(x)\right] = \mu_0(x)$$

$$\Sigma((\ell, x), (m, x')) = \text{Cov}(g(x) + \delta_\ell(x), g(x') + \delta_m(x')) = \Sigma_0(x, x') + \mathbb{1}_{\ell,m} \cdot \Sigma_\ell(x, x'),$$

where $\mathbb{1}_{\ell,m}$ denotes Kronecker's delta, and where we have used independence of $\delta_\ell, \delta_m$, and $g$. We refer the reader to `https://github.com/misoKG/` for an illustration of the model.

This generative model draws model discrepancies $\delta_\ell$ independently across IS. This is appropriate when IS are different in kind and share no relationship except that they model a common objective. In the supplement (Sect. B) we generalize this generative model to model correlation between model discrepancies, which is appropriate when IS can be partitioned into groups, such that IS within one group tend to agree more amongst themselves than they do with IS in other groups. Sect. B also discusses the estimation of the hyperparameters in $\mu_0$ and $\Sigma_\ell$.

## 3.2 Acquisition Function

Our optimization algorithm proceeds in rounds, selecting a design $x \in \mathcal{D}$ and an information source $\ell \in [M]_0$ in each. The value of the information obtained by sampling IS $\ell$ at $x$ is the expected gain in the quality of the best design that can be selected using the available information. That is, this value is the difference in the expected quality of the estimated optimum before and after the sample. We then normalize this expected gain by the cost $c_\ell(x)$ associated with the respective query, and sample the IS and design with the largest normalized gain. Without normalization we would always query the true objective, since no other IS provides more information about $g$ than $g$ itself.

We formalize this idea. Suppose that we have already sampled $n$ points $X_n$ and made the observations $Y_n$. Denote by $\mathbb{E}_n$ the expected value according to the posterior distribution given $X_n, Y_n$, and let $\mu^{(n)}(\ell, x) := \mathbb{E}_n[f(\ell, x)]$. The best *expected* objective value across the designs, as estimated by our statistical model, is $\max_{x' \in \mathcal{D}} \mu^{(n)}(0, x')$. Similarly, if we take an additional sample of IS $\ell^{(n+1)}$ at design $x^{(n+1)}$ and compute our new posterior mean, the new best expected objective value across the designs is $\max_{x' \in \mathcal{D}} \mu^{(n+1)}(0, x')$, whose distribution depends on what IS we sample, and where sample it. Thus, the expected value of sampling at $(\ell, x)$ normalized by the cost is

$$\text{MKG}^n(\ell, x) = \mathbb{E}_n \left[ \frac{\max_{x' \in \mathcal{D}} \mu^{(n+1)}(0, x') - \max_{x' \in \mathcal{D}} \mu^{(n)}(0, x')}{c_\ell(x)} \,\Big|\, \ell^{(n+1)} = \ell, x^{(n+1)} = x \right],$$
(1)

which we refer to as the `misoKG` factor of the pair $(\ell, x)$. The `misoKG` policy then samples at the pair $(\ell, x)$ that maximizes $\text{MKG}^n(\ell, x)$, i.e., $(\ell^{(n+1)}, x^{(n+1)}) \in \text{argmax}_{\ell \in [M]_0, x \in \mathcal{D}} \text{MKG}^n(\ell, x)$, which is a nested optimization problem.

To make this nested optimization problem tractable, we first replace the search domain $\mathcal{D}$ in Eq. (1) by a discrete set $\mathcal{A} \subset \mathcal{D}$ of points, for example selected by a Latin Hypercube design. We may then compute $\text{MKG}^n(\ell, x)$ exactly. Toward that end, note that

$$\mathbb{E}_n \left[ \max_{x' \in \mathcal{A}} \mu^{(n+1)}(0, x') \,\Big|\, \ell^{(n+1)} = \ell, x^{(n+1)} = x \right]$$
$$= \mathbb{E}_n \left[ \max_{x' \in \mathcal{A}} \{ \mu^{(n)}(0, x') + \bar{\sigma}_{x'}^n(\ell, x) \cdot Z \} \,\Big|\, \ell^{(n+1)} = \ell, x^{(n+1)} = x \right], \quad (2)$$

where $Z \sim \mathcal{N}(0, 1)$ and $\bar{\sigma}_{x'}^n(\ell, x) = \Sigma^n((0, x'), (\ell, x)) / [\lambda_\ell(x) + \Sigma^n((\ell, x), (\ell, x))]^{\frac{1}{2}}$. Here $\Sigma^n$ is the posterior covariance matrix of $f$ given $X_n, Y_n$.

We parallelize the computation of $\text{MKG}^n(\ell, x)$ for fixed $\ell, x$, enabling it to utilize multiple cores. Then $(\ell^{(n+1)}, x^{(n+1)})$ is obtained by computing $\text{MKG}^n(\ell, x)$ for all $(\ell, x) \in [M]_0 \times \mathcal{A}$, a task that can be parallelized over multiple machines in a cluster. We begin by sorting the points in $\mathcal{A}$ in parallel by increasing value of $\bar{\sigma}_{x'}^n(\ell, x)$ (for fixed $\ell, x$). Thereby we remove some points easily identified as dominated. A point $x_j$ is *dominated* if $\max_i \mu^{(n)}(0, x_i) + \bar{\sigma}_{x_i}^n(\ell, x) Z$ is unchanged for all $Z$ if the maximum is taken excluding $x_j$. Note that a point $x_j$ is dominated by $x_k$ if $\bar{\sigma}_{x_j}^n(\ell, x) = \bar{\sigma}_{x_k}^n(\ell, x)$ and $\mu^{(n)}(0, x_j) \le \mu^{(n)}(0, x_k)$, since $x_k$ has a higher expected value than $x_j$ for any realization of $Z$. Let $S$ be the sorted sequence without such dominated points. We will remove more dominated points later.

Since $c_\ell(x)$ is a constant for fixed $\ell, x$, we may express the conditional expectation in Eq. (1) as $\mathbb{E}_n \left[ \frac{\max_i \{a_i + b_i Z\} - \max_i a_i}{c_\ell(x)} \right] = \frac{\mathbb{E}_n[\max_i \{a_i + b_i Z\} - \max_i a_i]}{c_\ell(x)}$, where $a_i = \mu^{(n)}(0, x_i)$ and $b_i = \bar{\sigma}_{x_i}^n(\ell, x)$ for $x_i \in S$. We split $S$ into consecutive sequences $S_1, S_2, \ldots, S_C$, where $C$ is the number of cores used for computing $\text{MKG}^n(\ell, x)$ and $S_i, S_{i+1}$ overlap in one element: that is, for $S_j =$

$\{x_{j_1}, \ldots, x_{j_k}\}$, $x_{(j-1)_k} = x_{j_1}$ and $x_{j_k} = x_{(j+1)_1}$ hold. Each $x_{j_i} \in S_j$ specifies a linear function $a_{j_i} + b_{j_i} Z$ (ordered by increasing slopes in $S$). We are interested in the realizations of $Z$ for which $a_{j_i} + b_{j_i} Z \geq a_{i'} + b_{i'} Z$ for any $i'$ and hence compute the intersections of these functions. The functions for $x_{j_i}$ and $x_{j_{i+1}}$ intersect in $d_{j_i} = (a_{j_i} - a_{j_{i+1}})/(b_{j_{i+1}} - b_{j_i})$. Observe if $d_{j_i} \leq d_{j_{i-1}}$, then $a_{j_i} + b_{j_i} Z \leq \max\{a_{j_{i-1}} + b_{j_{i-1}} Z, a_{j_{i+1}} + b_{j_{i+1}} Z\}$ for all $Z$: $x_{j_i}$ is dominated and hence dropped from $S_j$. In this case we compute the intersection of the affine functions associated with $x_{j-1}$ and $x_{j+1}$ and iterate the process.

Points in $S_j$ may be dominated by the rightmost (non-dominated) point in $S_{j-1}$. Thus, we compute the intersection of the rightmost point of $S_{j-1}$ and the leftmost point of $S_j$, iteratively dropping all dominated points of $S_j$. If all points of $S_j$ are dominated, we continue the scan with $S_{j+1}$ and so on. Observe that we may stop this scan once there is a point that is not dominated, since the points in any sequence $S_j$ have non-decreasing $d$-values. If some of the remaining points in $S_j$ are dominated by a point in $S_{j'}$ with $j' < j - 1$, then this will be determined when the scan initiated by $S_{j'}$ reaches $S_j$. Subsequently, we check the other direction, i.e. whether $x_{j_1}$ dominates elements of $S_{j-1}$, starting with the rightmost element of $S_{j-1}$. These checks for dominance are performed in parallel for neighboring sequences.

[8] showed how to compute *sequentially* the expected maximum of a collection of affine functions. In particular, their Eq. (14) [8, p. 605] gives $\mathbb{E}_n \left[ \max_i \{a_i + b_i Z\} - \max_i a_i \right] = \sum_{j=1}^{C} \sum_{h=1}^{k-1} (b_{j_{h+1}} - b_{j_h}) u(-|d_{j_h}|)$, where $u$ is defined as $u(z) = z\Phi(z) + \phi(z)$ for the CDF and PDF of the normal distribution. We compute the inner sums simultaneously; the computation of the outer sum could be parallelized by recursively adding pairs of inner sums, although we do not do so to avoid communication overhead. We summarize the parallel algorithm below.

**The Parallel Algorithm to compute $(\ell^{(n+1)}, x^{(n+1)})$:**

1. Scatter the pairs $(\ell, x) \in [M]_0 \times \mathcal{A}$ among the machines.

2. Each computes $\text{MKG}^n(\ell, x)$ for its pairs. To compute $\text{MKG}^n(\ell, x)$ in parallel:

a. Sort the points in $\mathcal{A}$ by ascending $\bar{\sigma}_{x'}^n(\ell, x)$ in parallel, thereby removing dominated points. Let $S$ be the sorted sequence.

b. Split $S$ into sequences $S_1, \ldots, S_C$, where $C$ is the number of cores used to compute $\text{MKG}^n(\ell, x)$. Each core computes $\sum_{x_i \in S_C} (b_{i+1} - b_i) u(-|d_i|)$ in parallel, then the partial sums are added to obtain $\mathbb{E}_n \left[ \max_i \{a_i + b_i Z\} - \max_i a_i \right]$.

3. Determine $(\ell^{(n+1)}, x^{(n+1)}) \in \text{argmax}_{\ell \in [M]_0, x \in \mathcal{D}} \text{MKG}^n(\ell, x)$ in parallel.

### 3.3 Summary of the `misoKG` Algorithm.

1. Using samples from all information sources, estimate hyperparameters of the Gaussian process prior as described in Sect. B.2.
   Then calculate the posterior on $f$ based on the prior and samples.

2. Until the budget for samples is exhausted do:
   Determine the information source $\ell \in [M]_0$ and the design $x \in \mathcal{D}$ that maximize the `misoKG` factor proposed in Eq. (1) and observe IS $\ell(x)$.
   Update the posterior distribution with the new observation.

3. Return $\text{argmax}_{x' \in \mathcal{A}} \mu^{(n)}(0, x')$.

### 3.4 Provable Performance Guarantees.

The `misoKG` chooses an IS and an $x$ such that the expected gain normalized by the query cost is maximized. Thus, `misoKG` is *one-step Bayes optimal* in this respect, by construction.

We establish an *additive bound* on the difference between `misoKG`'s solution and the unknown optimum, as the number of queries $N \to \infty$. For this argument we suppose that $\mu(\ell, x) = 0 \; \forall \ell, x$ and $\Sigma_0$ is either the squared exponential kernel or a four times differentiable Matérn kernel. Moreover, let $x^{\text{OPT}} \in \text{argmax}_{x' \in \mathcal{D}} f(0, x')$, and $d = \max_{x' \in \mathcal{D}} \min_{x'' \in \mathcal{A}} \text{dist}(x', x'')$.

**Theorem 1.** *Let $x_N^* \in \mathcal{A}$ be the point that* `misoKG` *recommends in iteration $N$. For each $p \in [0, 1)$ there is a constant $K_p$ such that with probability $p$*

$$\lim_{N \to \infty} f(0, x_N^*) \geq f(0, x^{\mathrm{OPT}}) - K_p \cdot d.$$

We point out that Frazier, Powell, and Dayanik [8] showed in their seminal work an analogous result for the case of a single information source with uniform query cost (Theorem 4 in [8]).

In Sect. A we prove the statement for the MISO setting that allows multiple information sources that each have query costs $c_\ell(x)$ varying over the search domain $\mathcal{D}$. This proof is simple and short. Also note that Theorem 3 establishes *consistency* of `misoKG` for the special case that $\mathcal{D}$ is finite, since then $d = 0$. Interestingly, we can compute $K_p$ given $\Sigma$ and $p$. Therefore, we can control the additive error $K_p \cdot d$ by increasing the density of $\mathcal{A}$, leveraging the scalability of our parallel algorithm.

## 4 Numerical Experiments

We now compare `misoKG` to other state-of-the-art MISO algorithms. We implemented `misoKG`'s statistical model and acquisition function in `Python 2.7` and `C++` leveraging functionality from the `Metrics Optimization Engine` [23]. We used a gradient-based optimizer [28] that first finds an optimizer via multiple restarts for each *IS* $\ell$ separately and then picks $(\ell^{(n+1)}, x^{(n+1)})$ with maximum `misoKG` factor among these. An implementation of our method is available at `https://github.com/misoKG/`.

We compare to `misoEI` of Lam et al. [18] and to `MTBO+`, an improved version of Multi-Task Bayesian Optimization proposed by Swersky et al. [34]. Following a recommendation of Snoek 2016, our implementation of `MTBO+` uses an improved formulation of the acquisition function given by Hernández-Lobato et al. [12], Snoek and et al. [31], but otherwise is identical to `MTBO`; in particular, it uses the statistical model of [34]. Sect. E provides detailed descriptions of these algorithms.

**Experimental Setup.** We conduct experiments on the following test problems: (1) the 2-dimensional Rosenbrock function modified to fit the MISO setting by Lam et al. [18]; (2) a MISO benchmark proposed by Swersky et al. [34] in which we optimize the 4 hyperparameters of a machine learning algorithm, using a small, related set of smaller images as cheap IS; (3) an assemble-to-order problem from Hong and Nelson [13] in which we optimize an 8-dimensional target stock vector to maximize the expected daily profit of a company as estimated by a simulator.

In MISO settings the amount of initial data that one can use to inform the methods about each information source is typically dictated by the application, in particular by resource constraints and the availability of the respective source. In our experiments all methods were given *identical initial datasets* for each information source in every replication; these sets were drawn randomly via Latin Hypercube designs. For the sake of simplicity, we provided the same number of points for each IS, set to 2.5 points per dimension of the design space $\mathcal{D}$. Regarding the kernel and mean function, `MTBO+` uses the settings provided in [31]. The other algorithms used the squared exponential kernel and a constant mean function set to the average of a random sample.

We report the "gain" over the best initial solution, that is the true objective value of the respective design that a method would return at each iteration minus the best value in the initial data set. If the true objective value is not known for a given design, we report the value obtained from the information source of highest fidelity. This gain is plotted as a function of the *total cost*, that is the cumulative cost for invoking the information sources plus the fixed cost for the initial data; this metric naturally generalizes the number of function evaluations prevalent in Bayesian optimization. Note that the computational overhead of choosing the next information source and sample is omitted, as it is negligible compared to invoking an information source in real-world applications. Error bars are shown at the mean $\pm$ 2 standard errors averaged over at least 100 runs of each algorithm. For deterministic sources a jitter of $10^{-6}$ is added to avoid numerical issues during matrix inversion.

### 4.1 The Rosenbrock Benchmarks

We consider the design space $\mathcal{D} = [-2, 2]^2$, and $M = 2$ information sources. IS 0 is the Rosenbrock function $g(\boldsymbol{x}) = (1 - x_1)^2 + 100 \cdot (x_2 - x_1^2)^2$ plus optional Gaussian noise $u \cdot \varepsilon$. IS 1 returns

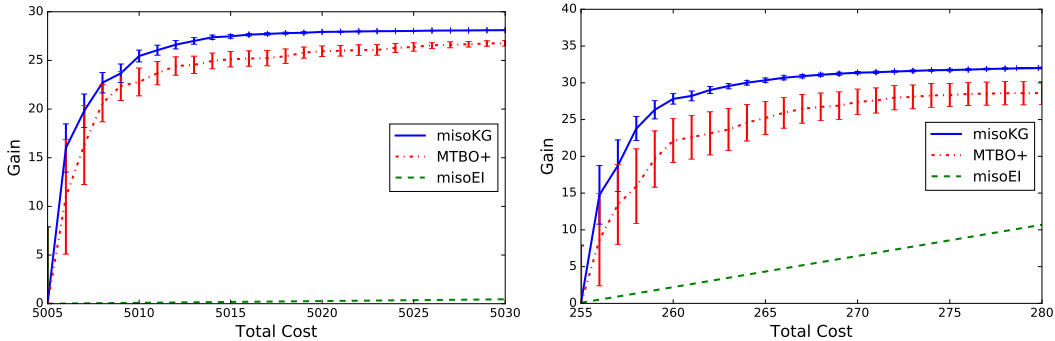

Figure 1: (l) The Rosenbrock benchmark with the parameter setting of [18]: `misoKG` offers an excellent gain-to-cost ratio and outperforms its competitors substantially. (r) The Rosenbrock benchmark with the alternative setup.

$g(\boldsymbol{x}) + v \cdot \sin(10 \cdot x_1 + 5 \cdot x_2)$, where the additional oscillatory component serves as model discrepancy. We assume a cost of 1000 for each query to IS 0 and a cost of 1 for IS 1.

Since all methods converged to good solutions within few queries, we investigate the ratio of gain to cost: Fig. 1 (l) displays the gain of each method over the best initial solution as a function of the total cost inflicted by querying information sources. The new method `misoKG` offers a significantly better gain per unit cost and finds an almost optimal solution typically within $5 - 10$ samples. Interestingly, `misoKG` relies only on cheap samples, proving its ability to successfully handle uncertainty. `MTBO+`, on the other hand, struggles initially but then eventually obtains a near-optimal solution, too. To this end, it makes usually one or two queries of the expensive truth source after about 40 steps. `misoEI` shows a odd behavior: it takes several queries, one of them to IS 0, before it improves over the best initial design for the first time. Then it jumps to a very good solution and subsequently samples only the cheap IS.

For the second setup, we set $u = 1$, $v = 2$, and suppose for IS 0 uniform noise of $\lambda_0(x) = 1$ and query cost $c_0(x) = 50$. Now the difference in costs is much smaller, while the variance is considerably bigger. The results are displayed in Fig. 1 (r): as for the first configuration, `misoKG` outperforms the other methods from the start. Interestingly, `misoEI`'s performance is drastically decreased compared to the first setup, since it only queries the expensive truth. Looking closer, we see that `misoKG` initially queries only the cheap information source IS 1 until it comes close to an optimal value after about five samples. It starts to query IS 0 occasionally later.

## 4.2 The Image Classification Benchmark

This classification problem was introduced by Swersky et al. [34] to demonstrate that `MTBO` can reduce the cost of hyperparameter optimization by leveraging a small dataset as information source. The goal is to optimize four hyperparameters of the logistic regression algorithm [36] using a stochastic gradient method with mini-batches (the learning rate, the L2-regularization parameter, the batch size, and the number of epochs) to minimize the classification error on the MNIST dataset [21]. This dataset contains 70,000 images of handwritten digits: each image has 784 pixels. IS 1 uses the USPS dataset [38] of about 9000 images with 256 pixels each. The query costs are 4.5 for IS 1 and 43.69 for IS 0. A closer examination shows that IS 1 is subject to considerable bias with respect to IS 0, making it a challenge for MISO algorithms.

Fig.2 (l) summarizes performance: initially, `misoKG` and `MTBO+` are on par. Both clearly outperform `misoEI` that therefore was stopped after 50 iterations. `misoKG` and `MTBO+` continued for 150 steps (with a lower number of replications). `misoKG` usually achieves an optimal test error of about 7.1% on the MNIST testset after about 80 queries, matching the classification performance of the best setting reported by Swersky et al. [34]. Moreover, `misoKG` achieves better solutions than `MTBO+` at the same costs. Note that the results in [34] show that `MTBO+` will also converge to the optimum eventually.

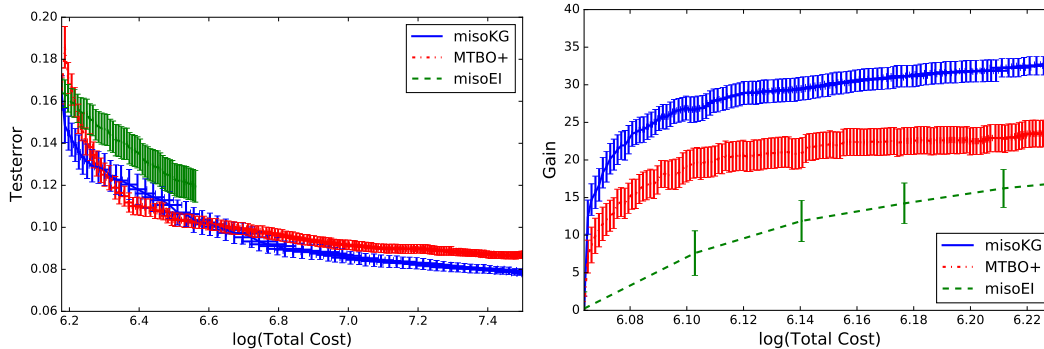

Figure 2: (l) The performance on the image classification benchmark of [34]. `misoKG` achieves better test errors after about 80 steps and converges to the global optimum. (r) `misoKG` outperforms the other algorithms on the assemble-to-order benchmark that has significant model discrepancy.

## 4.3 The Assemble-To-Order Benchmark

The assemble-to-order (ATO) benchmark is a reinforcement learning problem from a business application where the goal is to optimize an 8-dimensional target level vector over $[0, 20]^8$ (see Sect. G for details). We set up three information sources: IS 0 and 2 use the discrete event simulator of Xie et al. [42], whereas the cheapest source IS 1 invokes the implementation of Hong and Nelson. IS 0 models the truth.

The two simulators differ subtly in the model of the inventory system. However, the effect in estimated objective value is significant: on average the outputs of both simulators at the same target vector differ by about $5\%$ of the score of the global optimum, which is about $120$, whereas the largest observed bias out of $1000$ random samples was $31.8$. Thus, we are witnessing a significant model discrepancy.

Fig. 2 (r) summarizes the performances. `misoKG` outperforms the other algorithms from the start: `misoKG` averages at a gain of $26.1$, but inflicts only an average query cost of $54.6$ to the information sources. This is only $6.3\%$ of the query cost that `misoEI` requires to achieve a comparable score. Interestingly, `misoKG` and `MTBO+` utilize mostly the cheap biased IS, and therefore are able to obtain significantly better gain to cost ratios than `misoEI`. `misoKG`'s typically first calls IS 2 after about $60 - 80$ steps. In total, `misoKG` queries IS 2 about ten times within the first $150$ steps; in some replications `misoKG` makes one late call to IS 0 when it has already converged. Our interpretation is that `misoKG` exploits the cheap, biased IS 1 to zoom in on the global optimum and switches to the unbiased but noisy IS 2 to identify the optimal solution exactly. This is the expected (and desired) behavior for `misoKG` when the uncertainty of $f(0, x^*)$ is not expected to be reduced sufficiently by queries to IS 1. `MTBO+` trades off the gain versus cost differently: it queries IS 0 once or twice after $100$ steps and directs all other queries to IS 1, which might explain the observed lower performance. `misoEI`, which employs a two-step heuristic for trading off predicted gain and query cost, almost always chose to evaluate the most expensive IS.

## 5 Conclusion

We have presented a novel algorithm for MISO that uses a mean function and covariance matrix motivated by a MISO-specific generative model. We have proposed a novel acquisition function that extends the knowledge gradient to the MISO setting and comes with a fast parallel method for computing it. Moreover, we have provided a theoretical guarantee on the solution quality delivered by this algorithm, and demonstrated through numerical experiments that it improves significantly over the state of the art.

### Acknowledgments

This work was partially supported by NSF CAREER CMMI-1254298, NSF CMMI-1536895, NSF IIS-1247696, AFOSR FA9550-12-1-0200, AFOSR FA9550-15-1-0038, and AFOSR FA9550-16-1-0046.

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
