[Supplementary Material]

# Multi-Information Source Optimization

**Matthias Poloczek**      **Jialei Wang**      **Peter I. Frazier**

*Supplementary Material*

## A    Provable Performance Guarantees

First note that the `misoKG` chooses an IS and an $x$ such that the conditional expectation of the objective value of the recommendation is maximized per unit cost of the query. Let $N$ denote the number of queries that the policy is allowed to make.

**Proposition 1.** `misoKG` *achieves an optimal information gain per unit cost (with respect to $\mathcal{A}$) for* $N = 1$.

The proposition follows immediately by definition of the `misoKG` factor $\text{MKG}^n(\cdot, \cdot)$ in Eq. (1) on p. 4.

Next we establish an *additive* bound on the loss of the solution obtained by the algorithm with respect to the unknown optimum, as the number of queries $N \to \infty$. For the sake of convenience, we restate definitions given in Sect. 3.4. In what follows suppose that $\mu(\ell, x)=0 \;\forall \ell, x$ and $\Sigma_0$ is an at least four times differentiable Matérn kernel, e.g., the popular squared exponential kernel. Suppose that $f$ is drawn from the prior, i.e., let $f \sim GP(\mu, \Sigma)$. Then the sample $f(0, \cdot)$ from the distribution over functions is itself twice continuously differentiable with probability one (e.g., see Ghosal and Roy [9, Theorem 5]). Moreover, let $x^{\text{OPT}} \in \text{argmax}_{x' \in \mathcal{D}} f(0, x')$ and $d = \max_{x' \in \mathcal{D}} \min_{x'' \in \mathcal{A}} \text{dist}(x', x'')$. That is, $d$ is the maximum distance of any point in the continuous domain $\mathcal{D}$ to its closest point in the discrete set $\mathcal{A}$. Then we have the following bound on the objective value obtained by `misoKG`.

**Theorem 2** (Theorem 1 restated). *Let $x_N^* \in \mathcal{A}$ be the point that `misoKG` recommends in iteration $N$. For each $p \in [0, 1)$ there is a constant $K_p$ such that with probability $p$*

$$\lim_{N \to \infty} f(0, x_N^*) \geq f(0, x^{\text{OPT}}) - K_p \cdot d.$$

First observe that if $f(0, \cdot)$ is twice continuously differentiable, then in particular the extrema of $\frac{\partial}{\partial x_i} f(0, \cdot)$ over $\mathcal{D}$ are bounded. Specifically, since the partial derivatives of $GP(\mu, \Sigma)$ with respect to $x_i$ are also GPs for our choice [27, Sect. 9.4], we can compute for every $p \in [0, 1)$ a constant $K_p$ such that $f(0, \cdot)$ is $K_p$-Lipschitz continuous on $\mathcal{D}$ with probability at least $p$. Then there is an $\bar{x} \in \mathcal{A}$ with

$$\text{dist}(\bar{x}, x^{\text{OPT}}) \leq d$$

and hence

$$f(0, x^{\text{OPT}}) - f(0, \bar{x}) \leq K_p \cdot d. \tag{3}$$

We need one more caveat. Recall that $x_N^*$ is the recommendation of `misoKG` after $N$ queries to information sources.

**Theorem 3.** *As $N \to \infty$, $\lim f(0, x_N^*) \geq f(0, \bar{x})$ a.s.*

This completes the proof of Theorem 1, since we already showed that Eq. (3) holds with probability $p$.

We point out that Frazier, Powell, and Dayanik [8] showed in their seminal work that Theorem 3 holds for the case of a single information source with uniform query cost (Theorem 4 in [8]).

We prove it for the MISO setting that allows multiple information sources that each have query costs $c_\ell(x)$ varying over the search domain $\mathcal{D}$. In addition, the proof that we present in the following is simple and short. Also note that Theorem 3 establishes *consistency* of `misoKG` for the special case that $\mathcal{D}$ is finite, since then $d = 0$ holds.

*Proof.* We will show that the `misoKG` factor of each pair $\ell, x$ goes to zero almost surely, as the number of queries $n \to \infty$. This will imply that the algorithm identifies an optimal design in $\mathcal{A}$, thereby proving the claim. We define a filtration $(\mathcal{F}^n)$, where $\mathcal{F}^n$ is the $\sigma$-algebra generated by the first $n$

queries to information sources, denoted by $X = \{(\ell^{(i)}, x^{(i)}) | 1 \leq i \leq n\}$, and the corresponding observations $Y$. $\mathbb{E}_n := \mathbb{E}[\cdot | \mathcal{F}^n]$ is the expectation taken with respect to $\mathcal{F}^n$. Recall that we defined

$$\mu^{(n)}(\ell, x) = \mathbb{E}_n \left[ f(\ell, x) \right]$$

$$\bar{\sigma}_{x'}^n(\ell, x) = \Sigma^n((0, x'), (\ell, x)) / \left[ \lambda_\ell(x) + \Sigma^n((\ell, x), (\ell, x)) \right]^{\frac{1}{2}},$$

and let

$$V_n(\ell, x, \ell', x') = \mathbb{E}_n \left[ f(\ell, x) \cdot f(\ell', x') \right]$$
$$= \Sigma^n((\ell', x'), (\ell, x)) + \mu^{(n)}(\ell, x) \cdot \mu^{(n)}(\ell', x')$$

for $\ell \in [M]_0$, $x \in \mathcal{A}$, and $n \in \mathbb{N}_0$.

**Lemma 1.** *Let $\ell, \ell' \in [M]_0$ and $x, x' \in \mathcal{D}$. The limits of the series $(\mu^{(n)}(\ell, x))_n$ and $(V_n(\ell, x, \ell', x'))_n$ exist. Denote them by $\mu^{(\infty)}(\ell, x)$ and $V_\infty(\ell, x, \ell', x')$ respectively. We have*

$$\lim_{n \to \infty} \mu^{(n)}(\ell, x) = \mu^{(\infty)}(\ell, x) \tag{4}$$

$$\lim_{n \to \infty} V_n(\ell, x, \ell', x') = V_\infty(\ell, x, \ell', x') \tag{5}$$

*almost surely. If $(\ell', x')$ is sampled infinitely often, then $\lim_{n \to \infty} V_n(\ell, x, \ell', x') = \mu^{(\infty)}(\ell, x) \cdot \mu^{(\infty)}(\ell', x')$ holds almost surely.*

*Proof.* $f(\ell, x)$ and $f(\ell, x) \cdot f(\ell', x')$ are integrable random variables for all $\ell, \ell' \in [M]_0$ and $x, x' \in \mathcal{D}$ by choice of $f$. Proposition 2.7 in [5] states that any sequence of conditional expectations of an integrable random variable under an increasing filtration is a uniformly integrable martingale. Thus, both sequences converge almost surely to their respective limit.

If $(\ell', x')$ is sampled infinitely often, then its posterior variance goes to zero, and $\mathbb{E}_n \left[ f(\ell, x) \cdot f(\ell', x') \right] \to \mu^{(\infty)}(\ell, x) \cdot \mu^{(\infty)}(\ell', x')$. $\qquad\square$

Let $\omega$ denote an arbitrary sample path and note that $\omega$ determines an observation for each query to an information source, as $n \to \infty$. Then there must be a $f(\ell', x')$ that is observed infinitely often on $\omega$. Recall the definition of $\text{MKG}^n(\ell, x)$ from Eq. (1) and (2):

$$\text{MKG}^n(\ell', x') =$$
$$\frac{\mathbb{E}_n \left[ \max_{x'' \in \mathcal{A}} \{ \mu^{(n)}(0, x'') + \bar{\sigma}_{x'}^n(\ell', x') \cdot Z \} - \max_{x'' \in \mathcal{A}} \mu^{(n)}(0, x'') \mid \ell^{(n+1)} = \ell', x^{(n+1)} = x' \right]}{c_{\ell'}(x')}$$
$$\tag{6}$$

where $Z$ is a standard normal random variable and $c_{\ell'}(x')$ is a positive constant. We study the asymptotic behavior of $\text{MKG}^n(\ell', x')$ for $n \to \infty$ as a function of $\mu^{(n)}(\cdot, \cdot)$ and $\bar{\sigma}_\cdot^n(\cdot, \cdot)$.

**Lemma 2.** *Let $\ell' \in [M]_0$, $x' \in \mathcal{D}$ and suppose that $x'$ is observed infinitely often at IS $\ell'$ on $\omega$. Then $\bar{\sigma}_{x''}^n(\ell', x') \to 0$ for every $x'' \in \mathcal{D}$ and $\text{MKG}^n(\ell', x') \to 0$ almost surely, as $n \to \infty$.*

*Proof.* Lemma 1 implies that

$$\Sigma^n((\ell, x), (\ell', x')) = \mathbb{E}_n \left[ f(\ell, x) \cdot f(\ell', x') \right] - \mu^{(n)}(\ell, x) \cdot \mu^{(n)}(\ell', x') \to 0,$$

as $n \to \infty$. First suppose $\lambda_{\ell'}(x') = 0$ and that $x'$ is sampled at $IS$ $\ell'$ for the first time in iteration $u$. The update rule for the posterior variance gives that $\Sigma^{u+1}((\ell', x'), (\ell', x')) = \Sigma^u((\ell', x'), (\ell', x')) - \frac{\Sigma^u((\ell', x'), (\ell', x'))^2}{\Sigma^u((\ell', x'), (\ell', x'))} = 0$. Then we have for any $(\ell, x)$, $\mathbb{E}_{u+1}[f(\ell, x) \cdot f(\ell', x')] = \mu^{(u+1)}(\ell, x) \cdot \mu^{(u+1)}(\ell', x')$, and hence $\Sigma^{u+1}((\ell, x), (\ell', x')) = 0$ and $\text{MKG}^n(\ell', x') = 0$ hold in iteration $u + 1$ and all subsequent iterations.

Now suppose $\lambda_{\ell'}(x') > 0$. Then we have in particular

$$\lim_{n \to \infty} \bar{\sigma}_{x''}^n(\ell', x') = \lim_{n \to \infty} \Sigma^n((0, x''), (\ell', x')) / \left[ \lambda_{\ell'}(x') + \Sigma^n((\ell', x'), (\ell', x')) \right]^{\frac{1}{2}}$$
$$= 0$$

for all $x'' \in \mathcal{A}$. Note that the denominator is strictly positive since $\lambda_{\ell'}(x') > 0$. Recall that $(\mu^{(n)}(\ell', x'))_n$ and $(\bar{\sigma}^n_{x''}(\ell', x'))_n$ are uniformly integrable (u.i.) families of random variables that converge a.s. to their respective limits $\mu^{(\infty)}(\cdot, \cdot)$ and $\bar{\sigma}^\infty_{x''}(\ell', x') = 0$. Thus,

$$\lim_{n \to \infty} \text{MKG}^n(\ell', x')$$
$$= \frac{\int_{-\infty}^{+\infty} \phi(Z) \cdot \max_{x'' \in \mathcal{A}} \{\mu^{(\infty)}(0, x'') + \bar{\sigma}^\infty_{x''}(\ell', x') \cdot Z\} dZ - \max_{x'' \in \mathcal{A}} \mu^{(n)}(0, x'')}{c_{\ell'}(x')}$$
$$= 0,$$

where we used that $(\bar{\sigma}^n_{x''}(\ell', x') \cdot Z)_n$ is u.i., since $(Z)_n$ is independent of $(\bar{\sigma}^n_{x''}(\ell', x'))_n$, the sum of u.i. random variables is u.i., and so is the maximum over a finite collection of u.i. random variables. Moreover, $c_{\ell'}(x')$ is constant. $\qquad \square$

Recall that `misoKG` picks $(\ell^{(n+1)}, x^{(n+1)}) \in \text{argmax}_{\ell, x} \text{MKG}^n(\ell, x)$ in each iteration $n$. Since $f(\ell', x')$ is sampled infinitely often (by choice of $\ell', x'$), $\text{MKG}^n(\ell, x) \to 0$ holds a.s. for all $\ell \in [M]_0$ and $x \in \mathcal{A}$.

**Lemma 3.** *If* $\lim_{n \to \infty} MKG^n(\ell, x) = 0$ *holds for all* $\ell, x$, *then* $\text{argmax}_{x \in \mathcal{A}} \mu^{(\infty)}(0, x) = \text{argmax}_{x \in \mathcal{A}} f(0, x)$ *holds a.s.*

*Proof.* Lemma 1 gives that $\lim_{n \to \infty} \Sigma^n((0, x), (0, x)) = \Sigma^\infty((0, x), (0, x))$ a.s. for all $x \in \mathcal{A}$. First note that a maximizer is known perfectly if the posterior variance $\Sigma^\infty((0, x), (0, x)) = 0$ for all $x \in \mathcal{A}$. Thus, define $X = \{x \in \mathcal{A} \mid \Sigma^\infty((0, x), (0, x)) > 0\}$ and let $\hat{x} \in X$. Then

$$\bar{\sigma}^\infty_{\hat{x}}(0, \hat{x}) = \Sigma^\infty((0, \hat{x}), (0, \hat{x})) / [\lambda_0(\hat{x}) + \Sigma^\infty((0, \hat{x}), (0, \hat{x})]^{\frac{1}{2}} > 0. \qquad (7)$$

Note that $\text{MKG}^\infty(0, \hat{x}) > 0$ if there are $x_1, x_2 \in \mathcal{A}$ with $\bar{\sigma}^\infty_{x_1}(0, \hat{x}) \neq \bar{\sigma}^\infty_{x_2}(0, \hat{x})$. The reason is that then there is a $Z_0$ such that w.l.o.g. for all $Z > Z_0$, $\mu^{(\infty)}(0, x_1) + \bar{\sigma}^\infty_{x_1}(0, \hat{x}) \cdot Z > \mu^{(\infty)}(0, x_2) + \bar{\sigma}^\infty_{x_2}(0, \hat{x}) \cdot Z$ (and vice versa for $Z < Z_0$), resulting in a strictly positive numerator of Eq. (6). Thus, $\text{MKG}^\infty(0, \hat{x}) = 0$ implies $\bar{\sigma}^\infty_{x''}(0, \hat{x}) = \bar{\sigma}^\infty_{\hat{x}}(0, \hat{x})$ for *all* $x'' \in \mathcal{A}$, which is equivalent to

$$\frac{\Sigma^\infty((0, x'''), (0, \hat{x}))}{[\lambda_0(\hat{x}) + \Sigma^\infty((0, \hat{x}), (0, \hat{x}))]^{\frac{1}{2}}} = \frac{\Sigma^\infty((0, x''), (0, \hat{x}))}{[\lambda_0(\hat{x}) + \Sigma^\infty((0, \hat{x}), (0, \hat{x}))]^{\frac{1}{2}}}.$$

for all $x'', x''' \in \mathcal{A}$. In particular, Eq. (7) implies $\lambda_0(\hat{x}) + \Sigma^\infty((0, \hat{x}), (0, \hat{x})) > 0$ and hence $\Sigma^\infty((0, x'''), (0, \hat{x})) = \Sigma^\infty((0, x''), (0, \hat{x}))$. Thus, the covariance matrix of the $\{f(0, x) \mid x \in \mathcal{A}\}$ is proportional to the all-ones matrix, and hence $f(0, x) - \mu^{(\infty)}(0, x)$ is a normal random variable that is constant across all $x \in \mathcal{A}$. Therefore, $\text{argmax}_{x \in \mathcal{A}} \mu^{(\infty)}(0, x) = \text{argmax}_{x \in \mathcal{A}} f(0, x)$ holds. $\qquad \square$

Thus, a maximizer of $f(0, \cdot)$ over $\mathcal{A}$ is perfectly known (but not necessarily its exact objective value). We point out that this scenario cannot occur if there is only a single IS, as then $\ell' = 0$ and $\bar{\sigma}^\infty_{x''}(\ell', x') = 0$ together imply $X = \emptyset$. $\qquad \square$

## B  The Model Revisited

### B.1  Correlated Model Discrepancies

Next we demonstrate that our approach is flexible and can easily be extended to scenarios where some of the information sources have correlated model discrepancies. This arises for hyperparameter tuning if the auxiliary tasks are formed from data that was collected in batches and thus is correlated over time.

In engineering sciences we witness this if some sources share a common modeling approach, as for example, if one set of sources for an airfoil modeling problem correspond to different discretizations of a PDE that models wing flutter, while another set provides various discretizations of another PDE that modeling airflow. Two information sources that solve the same PDE will be more correlated than two that solve different PDEs.

Additionally, experiments conducted in the same location are exposed to the same environmental conditions or singular events, thus the outputs of these experiments might deviate from the truth by more than independent measurement errors. Another important factor is humans involved in the lab work, as typically workers have received the comparable training and may have made similar experiences during previous joint projects, which influences their actions and decisions.

For example, let $P = \{P_1, \ldots, P_Q\}$ denote a partition of $[M]_0$ and define the function $k : [M]_0 \to [Q]$ that gives for each IS its corresponding partition in $P$. Then we suppose an independent Gaussian process $\varepsilon(k(\ell), x) \sim GP(\mu_{k(\ell)}, \Sigma_{k(\ell)})$ for each partition. (Note that in principle we could take this approach further to arbitrary sets of $[M]_0$. However, this comes at the expense of a larger number of hyperparameters that need to be estimated.) Again our approach is to incorporate all Gaussian processes into a single one with prior distribution $f \sim GP(\mu, \Sigma)$:[1] therefore, for all $\ell \in [M]_0$ and $x \in \mathcal{D}$ we define $f(\ell, x) = f(0, x) + \varepsilon(k(\ell), x) + \delta_\ell(x)$, where $f(0, x) = g(x)$ is the objective function that we want to optimize. Due to linearity of expectation, we have

$$
\begin{aligned}
\mu(\ell, x) &= \mathbb{E}\left[f(0, x) + \varepsilon(k(\ell), x) + \delta_\ell(x)\right] \\
&= \mathbb{E}\left[f(0, x)\right] + \mathbb{E}\left[\varepsilon(k(\ell), x)\right] + \mathbb{E}\left[\delta_\ell(x)\right] \\
&= \mu_0(x),
\end{aligned}
$$

since $\mathbb{E}\left[\varepsilon(k(\ell), x)\right] = \mathbb{E}\left[\delta_\ell(x)\right] = 0$. Recall that the indicator variable $\mathbb{1}_{\ell,m}$ denotes Kronecker's delta. Let $\ell, m \in [M]_0$ and $x, x' \in \mathcal{D}$, then we define the following composite covariance function $\Sigma$:

$$
\begin{aligned}
&\Sigma\left((\ell, x), (m, x')\right) \\
&= \mathrm{Cov}\left(f(0, x) + \varepsilon(k(\ell), x) + \delta_\ell(x), f(0, x')\right. \\
&\quad \left. + \varepsilon(k(m), x') + \delta_m(x')\right) \\
&= \mathrm{Cov}(f(0, x), f(0, x')) + \mathrm{Cov}(\varepsilon(k(\ell), x), \varepsilon(k(m), x')) \\
&\quad + \mathrm{Cov}(\delta_\ell(x), \delta_m(x')) \\
&= \Sigma_0(x, x') + \mathbb{1}_{k(\ell),k(m)} \cdot \Sigma_{k(\ell)}(x, x') + \mathbb{1}_{\ell,m} \cdot \Sigma_\ell(x, x').
\end{aligned}
$$

## B.2 Estimation of Hyperparameters

In this section we detail how to set the hyperparameters via *maximum a posteriori* (MAP) estimation and propose a specific prior that has proven its value in our application and thus is of interest in its own right.

In typical MISO scenarios little data is available, that is why we suggest MAP estimates that in our experience are more robust than maximum likelihood estimates (MLE) under these circumstances. However, we wish to point out that we observed essentially the same performances of the algorithms when conducting the Rosenbrock and Assemble-to-Order benchmarks with maximum likelihood estimates for the hyperparameters.

In what follows we use the notation introduced in Sect. 3.1. One would suppose that the functions $\mu_0(\cdot)$ and $\Sigma_\ell(\cdot, \cdot)$ with $\ell \in [M]_0$ belong to some parameterized class: for example, one might set $\mu_0(\cdot)$ and each $\lambda_\ell(\cdot)$ to constants, and suppose that $\Sigma_\ell$ each belong to the class of Matérn covariance kernels (cp. Sect. 4 for the choices used in the experimental evaluation). The hyperparameters are fit from data using *maximum a posteriori* (MAP) estimation; note that this approach ensures that covariances between information sources and the objective function are inferred from data.

For a Matérn kernel we have to estimate $d + 1$ hyperparameters for each information source (see next subsection): $d$ length scales and the signal variance. We suppose a normal prior $\mathcal{N}\left(\mu_{\ell,i}, \sigma_{\ell,i}^2\right)$ for hyperparameter $\theta_{\ell,i}$ with $1 \le i \le d + 1$ and $\ell \in [M]_0$. Let $D \in \mathcal{D}$ be a set of points, for example chosen via a Latin Hypercube design, and evaluate every information source at all points in $D$. We estimate the hyperparameters for $f(0, \cdot)$ and the $\delta_\ell$ for $\ell \in [M]$, using the "observations" $\Delta_\ell = \{y(\ell, x) - y(0, x) \mid x \in D\}$ for the $\delta_\ell$. $y(\ell, x)$ is the observation including noise for design $x$ at IS $\ell$. The prior mean of the signal variance parameter of IS 0 is set to the variance of the observations at IS 0. The mean for the signal variance of IS $_s$ with $\ell \in [M]$ is obtained analogously using the "observations" in $\Delta_s$. If there is an estimate of the average observational noise variance then we

subtract it from the respective observations, exploiting the assumption that observational noise is independent. Regarding the means of the priors for length scales, we found it useful to set each prior mean to the length of the interval that the corresponding parameter is optimized over. For each hyperparameters $\theta_{\ell,i}$, we set the variance of the prior to $\sigma_{\ell,i}^2 = (\frac{\mu_{\ell,i}}{2})^2$, where $\mu_{\ell,i}$ is the mean of the prior.

### B.3 How to Express Beliefs on Fidelities of Information Sources

In many applications one has beliefs about the relative accuracies of information sources. One approach to explicitly encode these is to introduce a new coefficient $\alpha_\ell$ for each $\Sigma_\ell$ that typically would be fitted from data along with the other hyperparameters. But we may also set it at the discretion of a domain expert, which is particularly useful if none of the information sources is an unbiased estimator and we rely on regression to estimate the true objective. In case of the squared exponential kernel this coefficient is sometimes part of the formulation and referred to as "signal variance" (e.g., see [27, p. 19]). For the sake of completeness, we detail the effect for our model of uncorrelated information sources stated in Sect. 3.1. Recall that we suppose $f \sim GP(\mu, \Sigma)$ with a mean function $\mu$ and covariance kernel $\Sigma$, and observe that the introduction of the new coefficient $\alpha_\ell$ does not affect $\mu(\ell, x)$. But it changes $\Sigma((\ell, x), (m, x'))$ to

$$\Sigma((\ell, x), (m, x')) = \Sigma_0(x, x') + \mathbb{1}_{\ell,m} \cdot \alpha_\ell \cdot \Sigma_\ell(x, x').$$

We observe that setting $\alpha_\ell$ to a larger value results in a bigger uncertainty. The gist is that then samples from such an information source have less influence in the Gaussian process regression (e.g., see Eq. (A.6) on pp. 200 in [27]). It is instructive to consider the case that we observe a design $x$ at a noiseless and deterministic information source: then its observed output coincides with $f(\ell, x)$ (with zero variance). Our estimate $f(0, x)$ for $g(x)$, however, is a Gaussian random variable whose variance depends (in particular) on the uncertainty of the above information source as encoded in $\alpha_\ell$, since $\lambda_\ell(x) = 0$ holds.

## C  Limited Information Gain in Common Multi-Fidelity Models

In this section we show that the information that can be gained from sampling information sources of lower fidelities is limited for a common family of multi-fidelity models, used in [16, 14, 6, 20, 19]. They suppose that the IS form a strict hierarchy, where $IS$ 1 has lowest and $IS$ $M$ has highest fidelity, and denote the internal value of $IS$ $\ell$ at $x$ by $\hat{f}(\ell, x)$.

These works share the following two modeling assumptions:

1. Let $\hat{f}(\ell, x) = \rho_{\ell-1} \cdot \hat{f}(\ell - 1, x) + \hat{\delta}_\ell(x)$, where $\rho_\ell$ with $\ell \in [M]$ are known constants (e.g., estimated from data).

2. $\hat{\delta}_\ell$ and $\hat{\delta}_{\ell'}$ are pairwise independent Gaussian processes.

Kennedy and O'Hagan [16] state that the following "Markov property" holds for their model: $\text{Cov}\left(\hat{f}(\ell, x), \hat{f}(\ell - 1, x') \mid \hat{f}(\ell - 1, x)\right) = 0$ for all $x \neq x'$. It establishes that one cannot learn about $\hat{f}(\ell, x)$ by observing *any* $\hat{f}(\ell - 1, x')$ with $x' \neq x$, given that $\hat{f}(\ell - 1, x)$ is already known. We show that a slightly more general statement holds whenever the two above assumptions are met.

**Theorem 4.** *Let $\hat{f}(\ell, x) = \rho_{\ell-1} \cdot \hat{f}(\ell - 1, x) + \hat{\delta}_\ell(x)$, where $\rho_\ell$ with $\ell \in [M]$ are known constants, and $\delta_\ell, \delta_{\ell'}$ are pairwise independent Gaussian processes for all $\ell \neq \ell'$. If $\ell > \ell' > \ell''$, then*

$$Cov\left(\hat{f}(\ell, x), \hat{f}(\ell'', x') \mid \hat{f}(\ell', x)\right) = 0 \tag{8}$$

*holds for all $x \neq x'$.*

*Proof.* Expanding the covariance and $\hat{f}(\ell, x)$ gives

$$\text{Cov}\left(\hat{f}(\ell, x), \hat{f}(\ell'', x') \mid \hat{f}(\ell', x)\right)$$

$$= \text{Cov}\left(\hat{f}(\ell', x) \cdot \prod_{j=\ell'}^{\ell-1} \rho_j + \hat{\delta}_\ell(x) + \sum_{k=\ell'+1}^{\ell-1} \hat{\delta}_k(x) \cdot \prod_{j=k}^{\ell-1} \rho_j \ , \ \hat{f}(\ell'', x') \mid \hat{f}(\ell', x)\right)$$

$$= \text{Cov}\left(\hat{f}(\ell', x) \cdot \prod_{j=\ell'}^{\ell-1} \rho_j, \hat{f}(\ell'', x') \mid \hat{f}(\ell', x)\right) + \text{Cov}\left(\hat{\delta}_\ell(x), \hat{f}(\ell'', x') \mid \hat{f}(\ell', x)\right)$$

$$+ \sum_{k=\ell'+1}^{\ell-1} \text{Cov}\left(\hat{\delta}_k(x) \cdot \prod_{j=k}^{\ell-1} \rho_j \ , \ \hat{f}(\ell'', x') \mid \hat{f}(\ell', x)\right)$$

To see that the first summand is zero, we rewrite the covariance using $\text{Cov}(X, Y) = \mathbb{E}[X \cdot Y] - \mathbb{E}[X]\mathbb{E}[Y]$ (since $\hat{f}(\ell, x), \hat{f}(\ell'', x')$ are real-valued random variables with finite variance) and utilize that

$$\mathbb{E}\left[\hat{f}(\ell', x) \cdot \hat{f}(\ell'', x') \mid \hat{f}(\ell', x)\right] = \mathbb{E}\left[\hat{f}(\ell', x) \mid \hat{f}(\ell', x)\right] \cdot \mathbb{E}\left[\hat{f}(\ell'', x') \mid \hat{f}(\ell', x)\right]$$

holds.

For the second summand, again observe that all involved random variables are real-valued and have finite variance. Further, $\mathbb{E}\left[\hat{\delta}_\ell(x) \mid \hat{f}(\ell', x)\right] = \mathbb{E}\left[\hat{\delta}_\ell(x)\right]$, since $\hat{\delta}_\ell$ is independent of $\hat{\delta}_k$ for all $k < \ell$ and $\hat{f}(k, x)$ is a linear combination of $\hat{\delta}_1, \ldots, \hat{\delta}_k$ by definition. Let $p(\eta)$ be the conditional density of $\hat{\delta}_\ell(x)$ at $\eta$ given $\hat{f}(\ell', x)$ and $q(\nu)$ be the density of $\hat{f}(\ell'', x)$ at $\nu$ conditioned on $\hat{f}(\ell', x)$. Since $\hat{\delta}_\ell(x)$ and $\hat{f}(\ell'', x)$ are independent given $\hat{f}(\ell', x)$, their joint conditional density at $(\eta, \nu)$ equals $p(\eta) \cdot q(\nu)$, therefore

$$\mathbb{E}\left[\hat{\delta}_\ell(x) \cdot \hat{f}(\ell'', x') \mid \hat{f}(\ell', x)\right] = \int \int \eta \cdot \nu \, p(\eta) q(\nu) \, d\eta d\nu$$

$$= \int \eta \, p(\eta) \, d\eta \cdot \int \nu \, q(\nu) \, d\nu$$

$$= \mathbb{E}\left[\hat{\delta}_\ell(x)\right] \cdot \mathbb{E}\left[\hat{f}(\ell'', x') \mid \hat{f}(\ell', x)\right],$$

The remaining summands are zero by a similar argument, thereby proving the claim. $\qquad\square$

Note that the models proposed in Sect. 3.1 and Sect. B do not have this Markov property. Therefore, `misoKG` will be able to learn about $f(\ell, x)$ from additional queries to $IS$ $\ell''$. We regard this an important advantage in the context of multiple information source optimization, where information sources may have complementary strengths.

## D Extensions of the `misoKG` Algorithm

In this section we discuss extensions of the `misoKG` algorithm proposed in Sect. 3.2. We begin by discussing popular choices for the discretizations involved in the computation of `misoKG`. Then we show how the parallelization in Sect. 3.2 can be used to speed up gradient-based optimization of the acquisition function, e.g., used in [28, 25, 37]. We also extend this technique to the MISO setting. Finally, we show how a novel approach in [41] can be used to obtain a discretization-free formulation of `misoKG`.

**Varying Discrete Sets in the Inner and Outer Maximization Problem.** For the sake of simplicity, we employed only a single discretization $\mathcal{A}$ of the search domain $\mathcal{D}$ in Sect. 3.2. In practice, however, it is often advantageous to choose different discrete sets for the *inner* maximization problem $\mathbb{E}_n\left[\max_{x' \in \mathcal{A}_{\text{inner}}}\{\mu^{(n)}(0, x') + \bar{\sigma}_{x'}^n(\ell, x) \cdot Z\} \mid \ell^{(n+1)} = \ell, x^{(n+1)} = x\right]$ and the

*outer* maximization problem $\arg\max_{\ell\in[M]_0, x\in\mathcal{A}_{\text{outer}}} \mathrm{MKG}^n(\ell, x)$. A common approach is to pick new sets in each iteration $n$, for example drawn from the posterior on the location of the global optimizer: specifically, we draw $m$ samples $h^i$ with $1 \leq i \leq m$ from an approximate posterior on $f(0, \cdot)$ and then locate a global minimum $z^i$ of each $h^i$ using a gradient-based optimizer. Then we set $\mathcal{A}_n := \{z^1, \ldots, z^m\}$ in iteration $n$ (for more details, see Sect. 2.1 in Hernández-Lobato et al. [12] and Sect. 3.2 in Shah and Ghahramani [29]).

Another popular approach is to choose the discrete set as a grid of points with highest value under a weighted expected improvement criterion (e.g., see [11, 34]).

**Parallelization of Gradient-based Optimization of the `misoKG` Factor.** In practice it can be advantageous to compute the gradient of the acquisition function with respect to $x$ and then use a gradient-based optimizer to find the best $x$. This approach was proposed by Scott et al. [28] and later used in [25, 37]. All these articles consider a restricted scenario of a single information source without notion of costs.

We show how the parallelization in Sect. 3.2 can be used to speed up computations of their approaches. Moreover, we generalize the gradient-based optimization of the acquisition function to the MISO setting. Suppose that the functions $c_\ell(x)$ and $\lambda_\ell(x)$ are differentiable. For example, this is the case if they are estimated from data via Gaussian process regression with a constant mean function and a suitable, sufficiently smooth kernel [27, Sect. 9.4]. Now we use a result of [28] that $\frac{\partial}{\partial x}\mathbb{E}_n\left[\max_j\{a_j+b_j Z\} - \max_j a_j\right] = \sum_h(-\frac{\partial}{\partial x}b_{j_{h+1}}+\frac{\partial}{\partial x}b_{j_h})\phi(-|d_{j_h}|)$, where $\phi$ is the normal pdf. Note that the computation of $\frac{\partial}{\partial x}b_j$ follows the computation of $b_j$ in [8] (see [28, Sect. 5.2] for details), and hence can be parallelized analogously.

Therefore, we are able to compute $\frac{\partial}{\partial x}\mathrm{MKG}^n(\ell, x)$ in parallel for each $IS$ $\ell$ separately and then use a gradient-based optimizer to obtain a design $x_\ell^{(n+1)}$ with maximum value of information for each $\ell$. Note that multimodality of $\arg\max\mathrm{MKG}^n(\ell, x)$ is addressed by multiple restarts of the optimizer, e.g., starting from the points of a Latin hypercube design. Then the next sample pair $(\ell^{(n+1)}, x^{(n+1)})$ is obtained by comparing the best designs over all information sources and picking one with maximum `misoKG` factor.

**A Discretization-Free Formulation of `misoKG`.** Recently, Wu, Poloczek, Wilson, and Frazier [41] proposed a novel technique to avoid discretization in the inner maximization problem based on an envelope theorem [22]. If combined with the multi-start gradient-based approach for the outer maximization problem that we have described above, discretization can be avoided all together in the `misoKG` algorithm. We point out that the application of that technique is orthogonal to the contributions of this article.

# E   A Description of the MISO Benchmark Algorithms

The first benchmark method, `MTBO+`, is an improved version of Multi-Task Bayesian Optimization (`MTBO`) proposed by Swersky et al. [34]. It uses a cost-sensitive version of Entropy Search to select the next sample and information source: supposing a distribution over the location of the optima of the objective function, it maximizes in each iteration the reduction in differential entropy per query cost.

The algorithm requires a discretization $\mathcal{A}$ of $\mathcal{D}$. Let $P(x)$ be the probability that the optimum of $g$ is at $x\in\mathcal{A}$ conditioned on our previous observations, the hyperparameters, and $\mathcal{A}$, and let $H[P(x)]$ be the differential entropy of the corresponding distribution. Moreover, denote by $H[P_\ell^y(x)]$ the expected entropy of the distribution if we had sampled $x$ at IS $\ell$ and observed $y$. Then the cost-sensitive formulation of Entropy Search proposed in [34] is given by $\int\int (H[P(x)]-H[P_\ell^y(x)])\, p(y\,|\,\vec{f})p(\vec{f}\,|\,x,\ell)/c_\ell(x)\,\mathrm{d}y\mathrm{d}f$, where $p(\vec{f}\,|\,x,\ell)$ is the probability that the points in $\mathcal{A}$ take the values $\vec{f}$ conditioned on the hyperparameters and past observations, and $p(y\,|\,\vec{f})$ is the probability of observing $y$ when querying $x$ at IS $\ell$.

`MTBO` combines this acquisition function with a "multi-task" Gaussian process model. Their kernel is given by the tensor product $K_t \otimes K_x$, where $K_t$ (resp., $K_x$) denotes the covariance matrix of the tasks (resp., of the points), and hence is capable of exploiting correlations. Following a recommendation of

Figure 3: Evaluation of `misoKG`, `MF-GP-UCB`, `GP-UCB`, and `Random` on (l) the assemble-to-order benchmark, (m) the MISO Rosenbrock of [18], and (r) the alternative MISO Rosenbrock benchmark.

Snoek 2016, our implementation `MTBO+` uses an improved formulation of the acquisition function given by Hernández-Lobato et al. [12], Snoek and et al. [31], but otherwise is identical to `MTBO`; in particular, it uses the statistical model of Swersky et al. [34].

The other algorithm, `misoEI` of [18], was developed to solve MISO problems that involve model discrepancy and therefore is a good competing method to compare with.

It maintains a separate Gaussian process for each information source: to combine this knowledge, the corresponding posterior distributions are fused for each design via Winkler's method (1981) into a single intermediate surrogate, which is a normally distributed random variable. Then Lam et al. adapt the Expected Improvement (`EI`) acquisition function to select the design which is to be sampled next: for the sake of simplicity, assume that observations are noiseless and that $y^*$ is the objective value of a best sampled design. If $Y_x$ denotes a Gaussian random variable with the posterior distribution of the objective value for design $x$, then $\mathbb{E}[\max\{Y_x - y^*, 0\}]$ is the expected improvement for $x$, and the `EI` acquisition function selects an $x$ that maximizes this expectation. Based on this decision, the information source to invoke is chosen by a heuristic that aims at maximizing the `EI` per unit cost.

## F    Comparison to `MF-GP-UCB`, `GP-UCB`, and `Random`

We compare `misoKG` to the multi-fidelity method `MF-GP-UCB` of Kandasamy et al. [15] and to the single-fidelity methods `GP-UCB` of Srinivas et al. [33] and `Random` that picks points randomly. `misoKG` and `MF-GP-UCB` received the same number of initial samples from each IS; `GP-UCB` and `Random` obtain the same number of samples from IS 0 as `misoKG`. Fig. 3 summarizes the performances. We see that `misoKG` clearly outperforms the other algorithms on all benchmarks, leveraging cheap IS with great success. `MF-GP-UCB` on the other hand is not able to benefit from cheap approximations; it performs slightly worse than the single-fidelity methods, which suggests that the bias misleads the algorithm. This demonstrates that the performance of multi-fidelity methods degrades in the presence of model discrepancy. `GP-UCB` and `Random` only query the expensive IS 0 and hence cannot be competitive.

## G    Description of the Assemble-to-Order Benchmark

In the assemble-to-order (ATO) benchmark, a reinforcement learning problem from a business application, we are managing the inventory of a company that manufactures $m$ products. Each product is made from a selection from $n$ items, where we distinguish between key items and non-key items: a product can only be sold if all its key items are in stock. Non-key items are optional and increase the value. There is a target level for each item: the system automatically sends a replenishment order if the level drops below the target. The requested item is delivered after a random period. Since items in the inventory inflict holding cost, the goal is to find a target level vector that maximizes the expected profit per day (cp. Hong and Nelson [13] for details). The setting of Hong and Nelson supposes $m{=}5$ different products assembled from a subset of $n{=}8$ items, asking to optimize a 8-dimensional target vector $b \in [0, 20]^8$. We set up three information sources: IS 0 and IS 2 use the simulator of Xie et al. [42], whereas the cheapest source IS 1 invokes the implementation of Hong and Nelson. We assume that IS 0 models the truth. The IS differ in the number of replications: more replications increase the precision of the estimate but also the computational cost. IS 0 that models the truth has 500 replications, a noise variance of 0.056 and a cost of 17.1. IS 1 is the cheapest

IS with 10 replications, a noise of 2.944, and cost 0.5. IS 2 has 100 replications, noise 0.332, and cost 3.9. The observational noise and query cost were estimated from data, supposing for the sake of simplicity that both functions are constant over the domain.

The two simulators differ subtly in the model of the inventory system. However, the effect in estimated objective value is significant: on average the outputs of both simulators at the same target vector differ by about 5% of the score of the global optimum, which is about 120, whereas the largest observed bias out of 1000 random samples was 31.8. Thus, we are witnessing a significant model discrepancy.

## Footnotes

[1]For simplicity we reuse the notation from the first model to denote their pendants in this model.