[Reviews · NeurIPS 2017]

Reviewer 1



This paper deals with the important topic of optimization in cases where in addition to costly evaluations of the objective function, it is possible to evaluate cheaper approximations of it. This framework is referred to as MISO (multi-information source optimization) in the paper, where Bayesian optimization strategies relying on Gaussian process models are considered. An original MISO approach is presented, misoKG, that relies on an adaption of the Knowledge Gradient algorithm in multi-information source optimization settings. The method is shown to achieve very good results and to outperform considered competitors on three test cases. Overall, I am fond of the ideas and the research directions tackled by the paper, and I found the contributions principled and practically relevant. Given the potential benefits for the machine learning community and beyond, these contributions should definitely be published in a high-level venue such as NIPS. Yet, the paper could still be improved in several respects. Le me start by general remarks and then add minor comments. A lot of emphasis is put throughout the paper, and starting in the abstract, on the use of "a novel Gaussian process covariance kernel". In the introduction, "two innovations" are listed as 1) the kernel and 2) the acquisition function. I think that this is misleading, and to me the main contributions concern by far the acquisition function in the first place, and the model is more a modality of application. By the way, the proposed model is a quite simple particular case of the Linear Model of Corregionalization (which the authors mention), corresponding to an M*(M+1) matrix [1,I], i.e. a concatenation of a vector of ones with an identity matrix. Besides this, it also has quite some similarity with the Kennedy and O'Hagan model that was used by Forrester et al. in optimization, without neither the hierarchical aspect nor regression coefficients. In all, I have the feeling that using the model presented by the authors is indeed a clever thing to do, but to me it doesn't constitute a major contribution in itself. Not also that if misoKG is compared to several competitors, I did not see comparisons of the proposed model (in terms of fit) versus others. Also, what would prevent from using other criteria with this model or other models with the misoKG aquisition function? Other than that, let me formulate a few comments/questions regarding the main contributions presented in the third section. First, a finite set of points is introduced at each iteration for the calculation and the optimization of the acquisition function. I am wondering to what extent such a set is needed on the one hand for the calculation, on the other hand for the optimization, and finally if there are good reasons (beyond the argument of simplicity) why the same set should be used for both tasks. I conceive that optimizing over l and x at the same time is uneasy, but couldn't one optimize over D for every l and then compare the solutions obtained for the different l's? By the way, an approach was proposed in "Bayesian Optimization with Gradients" (arXiv:1703.04389) to avoid discretization in the calculation of a similar acquisition function; could such approach be used here? If yes but a discretization approach is preferred here, maybe a discussion on when considering discrete subsets anyway would make sense? On a different note, I was wondering to what extent the calculation tricks used around lines 158-172 differ from the ones presented in the PhD thesis of P. Frazier and related publications. Maybe a note about that would be a nice addition. Finally, I found the formulation of Theorem 1 slightly confusing. One speaks of a recommendation x_N^star in A and then take limit over N. So we have a sequence of points in A, right? Besides, it all relies on Corollary 1 from the appendix, but there the reader is referred to "the full version of the paper"...I do not have that in my possession, and as of now I found the proof of Corollary 1 somehow sketchy. I can understand that there are good reasons for that, but it is bit strange to comment on a result which proof is in appendix, which itself refers to a full version one doesn't have at hand! A few minor comments follow: * About the term "biased" (appearing in several places): I understand well that a zero-mean prior on the bias doesn't mean at all that there is no bias, but maybe manipulating means of biases without further discussion on the distinction between the bias and its model could induce confusion. Also, around lines 54-55, it is written that "...IS may be noisy, but are assumed to be unbiased" followed by "IS are truly unbiased in some applications" and "Thus, multi-fidelity methods can suffer limited applicability if we are only willing to use them when IS are truly unbiased...". As far as I understand, several methods including the ones based on the Kennedy and O'Hagan settings allow accounting for discrepances in the expectations of the different IS considered. So what is so different between multi-fidelity optimization and what is presented here regarding the account of possible "biases"? From a different perspective, other approaches like the one of "Quantile-Based Optimization of Noisy Computer Experiments with Tunable Precision" are indeed relying on responses with common expectation (at any given x). * In line 43, the "entropy" is the maximizer's entropy, right? * Still in line 43: why would the one-step optimality mean that the proposed approach should outperform entropy-based ones over a longer horizon? * About "From this literature, we leverage...", lines 70-72: as discussed above, it would be wothwhile to point out more explicitly what is new versus what is inherited from previous work regarding these computational methods. * In line 79 and in the bibliography: the correct last name is "Le Gratiet" (not "Gratiet"). * In line 84: without further precision, argmax is a set. So, the mentioned "best design" should be one element of this set? * In line 87: the "independence" is a conditional independence given g, I guess...? * In page 3 (several instances): not clear if the bias is g(.)-f(l,.) or delta_l(.)=f(l,.)-g(.)? * In line 159, the first max in the expectations should have braces or so around a_i + b_i Z. * In line 165: using c is a bit slippery as this letter is already used for the evaluation costs... * In "The Parallel Algorithm": strange ordering between 1) and 2) as 2) seems to describe a step within 1)...?

Reviewer 2



This paper proposes ideas on Bayesian optimization when the function to optimize is expensive to evaluate and a number of cheaper approximations are available. I think that the problem is interesting and worth investigating. The proposed model is essentially imposing a Gaussian process prior to the mismatch between the function and the approximations. In this respect, I'm not sure that the work proposes something really novel from the modeling perspective, given that these approaches have been investigated at large in the literature on quantification of uncertainty (see, e.g., [1,2]). However, the paper studies this in the context of optimization and the theorem proposed in the paper is interesting and it adds some solidity to the contribution. The paper proposes an ad-hoc objective function. I found it hard to read through page 4 where the paper illustrates the main idea of the paper. Perhaps some illustrations would be helpful here to better motivate the intuition behind the proposed objective. The experiments seem to indicate that the proposed objective function achieves better performance than the competitors. Overall the experimental section of the paper could be improved. The image classification benchmark is not so impressive - Neural Nets (even not convolutional), SVMs, Gaussian processes etc... achieve 98%+ accuracy on MNIST, and the optimized logistic regression achieves only ~93% accuracy. Maybe optimizing some of those models instead would make the results look better, while still demonstrating that the proposed method is superior to the competitors. In Sec 4.4 I find it surprising that "Random" performs as well as GP-UCB Sec 4.1 reports results on benchmarks, which I think is necessary and it's ok to have it there as a sanity check. References: [1] MC Kennedy and A O'Hagan. Bayesian calibration of computer models. Journal of the Royal Statistical Society Series B, 2001, vol. 63, issue 3, pages 425-464 [2] MC Kennedy, A O'Hagan; Predicting the output from a complex computer code when fast approximations are available. Biometrika 2000; 87 (1): 1-13.

Reviewer 3



1) You provide a description of how you pick the initial design, but you do not mention (as far as I could tell) how you pick the candidate set A in the experiments. You mention a hypercube design in the initial description, but is this to be understood as there being a fixed set of A being used throughout? What size? 2) How do you handle hyperparameters for the individual IS processes? 3) In computing the cost, you drop "dominated points". Won't that bias using the formula in line 173? (or step c in the algo). Also, is there anything preventing you from applying that formula immediately? (I'm guessing the instantiation of the sets S is done for computational reasons). 4) you say you use a gradient-based optimization algorithm, but isn't l a discrete variable over the sources? Is that handled by the multi-restart functionality and if yes, is it handled adequately? After rebuttal: 1,2,4) I think it would suit the paper to clarify the above to the extent possible. 3) I realize I was not familiar enough with the original work, and on inspection it appears to work as you say.